# Towards Learning Group-Equivariant Features for Domain Adaptive 3D Detection

**Sangyun Shin**[†]  **Yuhang He**[†]  **Madhu Vankadari**[†]  **Ta-Ying Cheng**[†]
**Qian Xie**[‡]  **Andrew Markham**[†]  **Niki Trigoni**[†]

[†]Department of Computer Science, University of Oxford, United Kingdom
[‡]School of Computer Science, University of Leeds, United Kingdom

 https://github.com/yunshin/GroupExp-DA.git
 sangyun.shin@cs.ox.ac.uk

## Abstract

The performance of 3D object detection in large outdoor point clouds deteriorates significantly in an unseen environment due to the inter-domain gap. To address these challenges, most existing methods for domain adaptation harness self-training schemes and attempt to bridge the gap by focusing on a single factor that causes the inter-domain gap, such as objects' sizes, shapes, and foreground density variation. However, the resulting adaptations suggest that there is still a substantial inter-domain gap left to be minimized. We argue that this is due to two limitations: 1) Biased pseudo-label collection from self-training. 2) Multiple factors jointly contributing to how the object is perceived in the unseen target domain. In this work, we propose a grouping-exploration strategy framework, Group Explorer Domain Adaptation (*GroupEXP-DA*), to addresses those two issues. Specifically, our grouping divides the available label sets into multiple clusters and ensures all of them have equal learning attention with the group-equivariant spatial feature, avoiding dominant types of objects causing imbalance problems. Moreover, grouping learns to divide objects by considering inherent factors in a data-driven manner, without considering each factor separately as existing works. On top of the group-equivariant spatial feature that selectively detects objects similar to the input group, we additionally introduce an explorative group update strategy that reduces the false negative detection in the target domain, further reducing the inter-domain gap. During inference, only the learned group features are necessary for making the group-equivariant spatial feature, placing our method as a simple add-on that can be applicable to most existing detectors. We show how each module contributes to substantially bridging the inter-domain gaps compared to existing works across large urban outdoor datasets such as NuScenes, Waymo, and KITTI.

## 1  Introduction

Learning 3D object detection [42, 12, 27] and segmentation [50, 11, 32, 10] in large outdoor point cloud scenes is of increasing importance due to its wide range of applications, such as autonomous driving [19] and Augmented Reality (AR) [14]. However, annotating such 3D data is expensive and labor-intensive. This limits the applicability and generalization of off-the-shelf models to diverse real-world applications. To mitigate the dependency on large-scale datasets with labels, studies for Domain Adaptation (DA) have gained considerable attention from the community [38, 25, 41, 46, 44, 43, 20, 17, 40, 8, 3, 31, 7]. In the context of 3D detection, DA is motivated by utilizing knowledge learned from a source domain to adapt to a target domain using a pseudo-label set.

38th Conference on Neural Information Processing Systems (NeurIPS 2024).

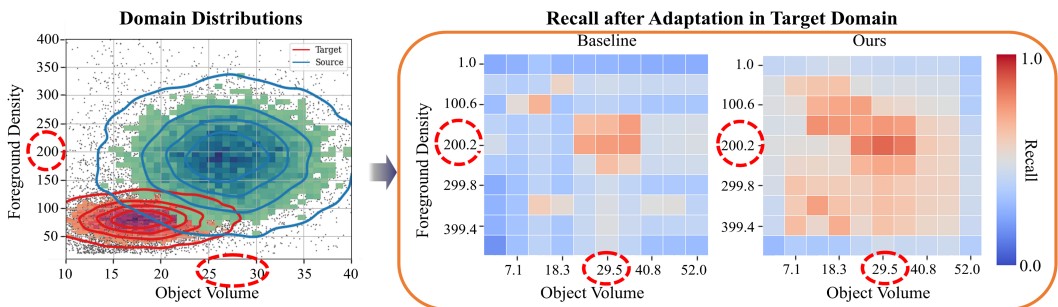

Figure 1: Several factors causing the inter-domain gap, such as the point density and object volume in NuScenes (Target) and Waymo (Source) datasets are illustrated with the fitted multivariate Gaussian distribution (left). The baseline [43] adaptation primarily detects objects having features near the mean of the distributions indicated by red circles. On the other hand, the proposed adaptation first groups objects and explores the target domain to reduce the false negative. The heatmaps show average recall (right). Objects with extreme sparsity are excluded for clear visualization.

Typically, in the widely used self-training scheme [44, 43], a detection model is pre-trained on the source domain and constructs an initial pseudo-label set for re-training in the target domain. The pseudo-label set progressively expands as more pseudo-labels are collected after each retraining. Here, the pseudo-label set consists of detections with high confidence scores. Based on the principle of self-training, recent existing works focus on specific factors that cause the inter-domain gap. These can be broadly categorized into domain variations in object sizes [38], density [8], or geometric structure [17, 20]. Despite good progress in bridging the inter-domain gap, two challenges remain unsolved: 1) Biased pseudo-labels due to the conservative collection strategy of the self-training scheme 2) Strict separation of multiple factors that jointly contribute to the creation of inter-domain gap. Typically, in one domain, a group of objects sharing common features outnumber other objects having different features, as addressed in existing works [38, 8, 17, 20]. While this may not cause a significant performance deterioration in the source domain, where the environment is similar, it could cause a bias under the self-training scheme. For example, detection with high confidence scores typically comes from objects belonging to dominant groups, as they are the ones that the detector learns the most of in the source domain. As shown in Figure 1 (a), the recall of objects in the target domain is significantly higher when the objects contain similar features as the dominant objects in the source domain, ignoring other objects. As a partial solution to this problem, existing works focus on a single factor only to address the inter-domain gap e.g. size or point cloud density. However, as Figure 1 shows, a single factor cannot explain all the domain variation as an object's appearance is a result of multiple factors jointly influencing the foreground points.

In this work, we address the aforementioned two issues of the current domain adaptive 3D detection. Our core intuition comes from the fact that the factors causing inter-domain gaps often already exist in the source domain to an extent. For example, the density of points often varies due to the distance and viewing angle of the sensor even inside a domain. Object sizes and shapes would also vary even in one single domain. Based on this observation, we aim to reduce the bias of the detection model to dominant objects by finding groups and evenly distributing the detector's attention during learning to all groups that would be otherwise largely neglected due to less dominant features. Nevertheless, finding optimal groups that represent the intra-domain gap is not straightforward because multiple factors jointly contribute to variations in objects' appearances. To address this issue, we introduce a data-driven grouping method that finds object groups with different characteristics. The groups are then progressively updated for adaptation, redistributing the available labels according to each group to learn the different characteristics found for each group in the target domain. Our contributions can be summarized as follows:

1. We introduce a new alternative approach for domain adaptation, **Group Expl**orer **D**omain **A**daptation, (***GroupExp-DA***), which reflects on the available labels in order to understand the target domain, bridging the inter-domain gap.

2. To ensure each object group receives equal attention for learning, we introduce the Group-Equivariant spatial feature, which is learned for selectively detecting objects similar to the input group, preventing dominant types of objects from causing imbalance problems.

3. To make the best use of the Group-Equivariant feature's selective detection ability depending on the input group, we propose an exploration strategy that encourages the groups to reflect the target domain by redistributing available labels, leading to fewer false negatives for the adaptation.

4. Extensive adaptation experiments on KITTI, Waymo, and NuScenes datasets show the effectiveness of our approach for bridging inter-domain gaps.

## 2 Related Work

**Point Cloud-based 3D Detection for Outdoor Scene.** Existing methods for 3D point cloud-based object detection can be divided into two main categories: point-based and voxel-based. Based on PointNet series [21, 24], point-based methods [23, 29, 29, 22, 45] propose to extract features from unordered and unstructured raw point-cloud directly. Voxel-based methods divide the unstructured point clouds into regular voxel grids and are fed into encoders either based on sparse convolution [42, 51, 12, 30, 26, 2, 9, 48] or Transformer [34, 16, 5, 37, 15]. The encoded spatial features, which are also called Bird's Eyes View (BEV) features, are then fed into Regional Proposal Networks (RPN) for the final detection. There are also a few methods that combine point- and voxel-based methods [27, 28, 39]. Currently, in terms of performance, voxel-based detectors dominate the point-cloud-based 3D detection for outdoor scenes. Therefore, we employ Second-IoU [42] and PointPillars [12] as the base detectors for our experiments, as they are most-widely used detectors that existing works are built on.

**Domain Adaptive Detection.** The target of domain adaptive 3D detection is to mitigate the inter-domain gap between the source and the target by focusing on several factors, such as object size [38, 25], point cloud deterioration [41], and the encoder separation for domains [46]. ST3D series [44, 43] propose a self-training scheme that progressively collects pseudo-label sets in the target domain for retraining. Built on self-training scheme, DA for 3D detection has been extensively studied to acquire higher quality pseudo-labels by addressing the inter-domain gap caused by geometric shape with prototype learning [20, 17], dense-to-sparse density variation using knowledge distillation [40], a general density variation with beam augmentation and knowledge distillation [8], cross-domain examination to measure the consistency of pseudo-labels [3] and focusing on specific architecture [49]. A considerable number of existing works focus on a single factor, such as density, shape, and size, to bridge the inter-domain gap. Instead, we attempt to see the inter-domain gap as a result of multiple factors combined and bridge the gap by finding inherent groups.

## 3 Method

### 3.1 Framework Overview

Following the self-training-based unsupervised domain adaptation scheme [44, 43] for 3D detection, we are given point clouds $X = X_s \cup X_t$ and labels $Y = Y_s \cup Y_t$ as the initial set. Here, $X_s$ and $Y_s$ are point could and box labels of the known source domain. On the other hand, $X_t$ and $Y_t$ are the point cloud and initial pseudo label set in the unlabeled target domain. $Y_t$ is collected by the detector trained only on the source domain using $X_s$ and $Y_s$. A label in $Y$ consists of seven parameters defining a 3D box with three parameters for center $(x, y, z)$, three parameters for size $(l, w, h)$, and one parameter for vertical rotation $\theta$. Our goal is to improve the detector's inter-domain adaptation by focusing on the pseudo-label collection. In particular, instead of treating all available objects in $Y$ equivalently, we aim to understand objects in $Y$ better by grouping.

Specifically, as depicted in Figure 2 and Alg. 1, we first extract foreground points from available boxes and learn to encode objects into the object descriptor from the points (Sec.3.2). The descriptors are then used to progressively group objects (Sec. 3.3). Our Group-Region Correlation (Sec. 3.4) takes the group features as input and fuses with RPN to selectively detect objects similar to the individual group.

### 3.2 Object Descriptor Extraction

Given a pair of point cloud $x \in \mathbb{R}^{n_p \times 3}$ and a label set $y \in \mathbb{R}^{n_b \times 7}$ consisting of $n_p$ points and $n_b$ boxes from $X$ and $Y$, respectively, this module aims to produce a learnable object descriptor $F_{obj} \in \mathbb{R}^{n_b \times d_{obj}}$ during training. First, foreground points, $P_{obj} = \{p_{obj}^k\}_{k=1}^{n_b}$ are extracted using $y$

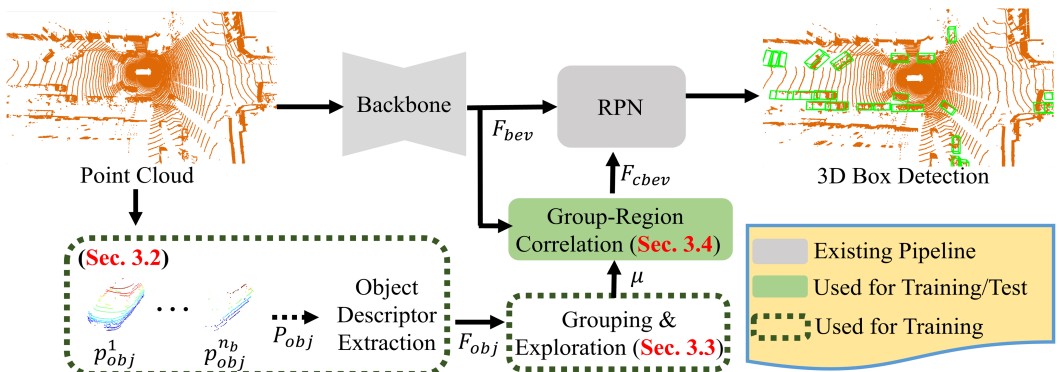

Figure 2: Overall pipeline of our proposed method. During training, we extract foreground points from existing 3D box labels and feed them to the Object Descriptor Extraction module to acquire object descriptors. The descriptors are used for grouping & exploration and fed into the Group-Correlation module to generate RPN to detect objects similar to each group.

from the input point cloud $x$. Our object descriptor extraction module then takes $P_{obj}$ as input and outputs object descriptors $F_{obj}$ using neural networks consisting of MLP and global max pooling, which are adopted from [21, 13]. The motivation behind the architectural choice is two-fold: (1) An arbitrary number of object points, $p_{obj}^k$, can be processed efficiently without involving comparably slow sampling techniques. (2) Global max pooling offers permutation invariant features, which helps $F_{obj}$ less prone to overfitting by certain permutations from viewing angles, *etc.* During training, $F_{obj}$ serves as the input for the progressive grouping process and is not used during inference.

## 3.3 Progressive Grouping

Determining similarities between objects is a complex problem as intra-object variations are created by various factors, such as density, shape, size, *etc.* To find groups that, when combined, explain this intra-object variation, we utilize Gaussian Mixture Model (GMM) based grouping for the following advantages: (1) GMM better captures the heterogeneity of data using only a few more parameters, such as covariance and weights, compared to the proximity-based methods. (2) The parameters that define groups can be efficiently updated with weighted linear combinations and are also differentiable for learning the groups in a data-driven manner. In the following sections, we explain the details of how the groups are initialized, determined for each sample, and updated.

**Initialization** Given all available pairs of scans and labels, X and Y, we first extract $F_{obj}$ from each pair and stack them. All of the stacked $\{F_{obj}^o\}_{o=1}^{n_{total}}$ are then used for the initialization. Here, $n_{total}$ stands for the total number of pairs in X and Y. Specifically, we initialize $n_g$ groups using K-Means clustering on $\{F_{obj}^o\}_{o=1}^{n_{total}}$, where each cluster forms a group. After this, Maximum Likelihood Estimation is performed for each group to acquire parameters, such as mean $\mu \in \mathbb{R}^{n_g \times d_{obj}}$, covariance $\sigma \in \mathbb{R}^{n_g \times d_{obj} \times d_{obj}}$, and weight $\phi \in \mathbb{R}^{n_g}$ that define a GMM-based group. This initialization is required only once before the training. In the following, all procedures are based on a single pair of scan and label, $x$ and $y$, which can be extended to batch-wise operation.

**Determining Group for New Sample** The probability $P^{k \to i}$ of $k$-th sample belonging to $i$-th group is estimated as :

$$P^{k \to i} = \frac{\phi^i \mathcal{N}(F_{obj}^k | \mu^i, \phi^i)}{\sum_{t=1}^{n_g} \phi^t \mathcal{N}(F_{obj}^k | \mu^t, \phi^t)}, \tag{1}$$

where $\mathcal{N}(|,)$ is the probability density function of multivariate Gaussian distribution that outputs the likelihood of a sample $F_{obj}^k$ given mean $\mu$ and covariance $\phi$. The group labels $G \in \mathbb{R}^{n_b}$ for all the $n_b$ samples in $F_{obj}$ are then acquired using $P$. That is, the group label $G^k$ for $k$-th sample is specifically determined as:

$$G^k = \arg\max_i P^{k \to i} \tag{2}$$

**Explorative Update during Training** Typically, pseudo-label sets from the target domain are incorporated into the existing label set from the source domain and considered as the same label set

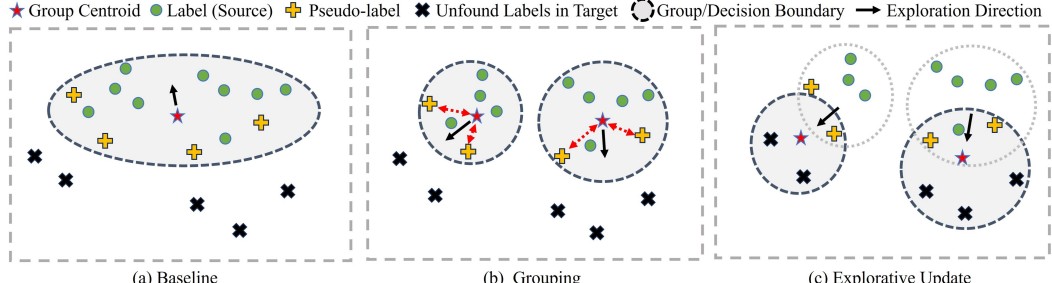

★ Group Centroid  ● Label (Source)  ➕ Pseudo-label  ✖ Unfound Labels in Target  ⬭ Group/Decision Boundary  ➝ Exploration Direction

(a) Baseline                    (b) Grouping                    (c) Explorative Update

Figure 3: Conceptual diagram comparing pseudo-label generation processes of (a) baseline [44, 43] with our (b) grouping followed by (c) the explorative update using pseudo-labels.

for training [44, 43], as shown in Figure 3 (a). However, we argue that for the domain adaptation, the pseudo-label set should be given more weight on its usage than the source label sets because they contain inherent features of the target domain, which can be used to understand the target domain in terms of objects' appearances. To address this, we introduce an explorative group update strategy using pseudo-labels. For $i$-th group, its mean $\hat{\mu}^i \in \mathbb{R}^{d_{obj}}$, covariance $\hat{\sigma}^i \in \mathbb{R}^{d_{obj} \times d_{obj}}$, and the group weight $\hat{\phi}^i$ are acquired as follows:

$$\hat{\mu}^i = \frac{1}{n_{b,i}} \sum_{k=1}^{n_{b,i}} F_{b,i}^k, \quad \hat{\sigma}^i = \frac{1}{n_{b,i}} \sum_{k=1}^{n_{b,i}} (F_{obj,i}^k - \hat{\mu}^i)(F_{obj,i}^k - \hat{\mu}^i)^T, \quad \hat{\phi}^i = \frac{n_{b,i}}{n_b}, \quad (3)$$

where $F_{obj,i}$ is subset of $F_{obj}$ that belong to $i$-th group using $G$ in Eqn. 2 as:

$$F_{obj,i} = \{F_{obj}^k | G^k = i\}, \quad (4)$$

and $n_{b,i}$ is the number of samples in $F_{obj,i}$. Similar to the update rule for prototype learning [20], each group parameters are updated with linear combination as:

$$\mu = \alpha\mu + (1-\alpha)\hat{\mu}, \quad \sigma = \alpha\sigma + (1-\alpha)\hat{\sigma}, \quad \phi = \alpha\phi + (1-\alpha)\hat{\phi}, \quad (5)$$

where $\alpha$ is a coefficient that affects how conservatively each parameter is updated. After the update, $\mu^i$ is considered as a representative of $i$-th group and utilized as a query input for generating group-equivariant spatial features. Accordingly, during training, the samples that belong to the $i$-th group are used as the foreground boxes. Intuitively, this process distributes source labels according to the groups found in the target domain so that the source labels are used for learning to find similar objects in the target domain, as shown in Figure 3 (b) and (c).

For training, to ensure that the groups learn to be distinctive enough, we adopt inter-group repel [17] based on the contrastive loss.

$$L_{rep} = \sum_{i=1}^{n_g-1} \sum_{j=i+1}^{n_g} max(0, cos(\mu^i, \mu^j)), \quad (6)$$

where $cos(,)$ is a function that calculates cosine-similarity between two inputs. In addition, to encourage similar features for group cohesion, intra-group attraction loss, $L_{att}$ is used:

$$L_{att} = \sum_{i=1}^{n_g} \sum_{k=1}^{n_b} (1 - cos(F_{obj}^k, \mu^i))\mathbb{1}[G^k = i], \quad (7)$$

where $\mathbb{1}[G^k = i]$ is an indicator function that is 1 if $G^k = i$ and 0 otherwise. During the inference, only the learned $\mu$ is necessary for the following Group-Region Correlation.

### 3.4 Group-Region Correlation

In a typical voxel-based 3D object detector's pipeline, voxelized points are fed into the backbone, which outputs the spatial feature $F_{bev} \in \mathbb{R}^{H \times W \times d_{bev}}$, as shown in Figure 2. $F_{bev}$ is then fed into RPN to make final box predictions. Given input group queries $\mu = \{\mu^i\}_{i=1}^{n_g}, \mu^i \in \mathbb{R}^{d_{obj}}$ and $F_{bev}$, Our Group-Region Correlation aims at producing the spatial features $F_{cbev} \in \mathbb{R}^{n_g \times H \times W \times d_{bev}}$ that

---
**Algorithm 1** Training Pipeline
---
**Require:** a point-cloud $x$, (pseudo) labels $y$, Group parameters $G$

 1: Extract obj. descriptors $F_{obj}$ using $x$ and $y$ (Sec 3.2)
 2: Determine groups of $F_{obj}$ and update $G$ (Sec 3.3)
 3: Update $G$ using $F_{obj}$ according to the determined groups (Sec 3.3)
 4: Calculate $L_{rep}$ and $L_{att}$ as in Sec 3.3
 5: Extract spatial feature $F_{bev}$ from the backbone (Sec 3.4)
 6: Make group equivariant features for each group using G and $F_{bev}$ (Sec 3.4)
 7: Predict boxes using shared RPN for each group equivariant feature
 8: Calculate $L_{det}$ as in Eq. 10.
 9: Calculate gradients using $L_{rep}$, $L_{att}$, and $L_{det}$.
10: Update all modules' weights using the gradients
---

are equivariant to the group query so that $F_{cbev}$ provide selective features for detecting objects similar to each group. In the following sections, we explain how each group and spatial feature are correlated and then used by RPN to detect the corresponding objects.

**Group Equivariant Spatial Feature** The aim of Group-Region Correlation module is to make spatial features $F_{bev}$ selectively attend to objects that are similar to the query group. The following RPN then detects only certain objects that are similar to the query group. To achieve this, we utilize cross-attention with $\mu$ as query and $F_{bev}$ as key and value to encourage features from $\mu$ and $F_{bev}$ to cross-attend to generate necessary features. Intuitively, $\mu$ is compared to $F_{bev}$ to find the object that is similar to each group. For $i$-th group, the attended group-equivariant spatial features $F_{cbev}^i \in \mathbb{R}^{H \times W \times d_{bev}}$ are acquired as:

$$F_{cbev}^i = cAttn(\mu^i, F_{bev}, F_{bev}), \tag{8}$$

where $cAttn(.)$ refers to the cross attention [36] that takes query, key, and value as input and outputs the cross-attended feature. Here, $\mu^i$ is the $i$-th query in $\mu$. The group-equivariant spatial features $F_{cbev}$ can then be fed into any existing RPN strcutures [42, 4, 12] to detect foreground objects for each group. The ground-truth boxes in $y$ that belong to the $i$-th group are utilized as the foreground boxes for $F_{cbev}^i$ to train the follwing RPN.

**Regional Proposal Network (RPN)** Following the general architecture of RPN [42, 12], our objectness and box regression heads take $F_{cbev} \in \mathbb{R}^{n_g \times H \times W \times D}$ as input and predict objectness scores $F_{cls} \in \mathbb{R}^{n_g \times H \times W \times 1}$ and box parameters $F_{box} \in \mathbb{R}^{n_g \times H \times W \times 7}$ to form 3D boxes on the dense spatial grid corresponding to $F_{cbev}$.

For the training of $i$-th group, standard training losses for RPN based detection, $L_{det}^i$, are applied as existing works [42, 12, 30, 26, 2, 9, 48]. Specifically, given the box labels $y^i$, $F_{cls}^i$ and $F_{box}^i$ are used for calculating first-stage box detection training loss, $L_{det1}$, as:

$$L_{det1}^i = L_{focal}^i + L_{box}^i, \tag{9}$$

where $L_{focal}$ stands for Focal Loss [18] and $L_{box}$ is box regression loss. Here, the foreground labels and regression targets for $L_{focal}$ and $L_{box}$ are calculated using $y^i$ depending on the individual base detectors' configurations. Similarly, for the second-stage box refinement training, $F_{cbev}$ is used with $F_{cls}$ and $F_{box}$ for RoI Pooling. Then, the pooled features are fed into classification and box regression head for refinement with architectures depending on the detectors to calculate the second stage loss $L_{det2}^i$. The detection loss for all groups, $L_{det}$, is then acquired by iterating over all groups as:

$$L_{det} = \frac{1}{n_g} \sum_{i=1}^{n_g} L_{det1}^i + L_{det2}^i. \tag{10}$$

## 3.5 Overall Training

Apart from $L_{rep}$ and $L_{att}$ for grouping, our overall training losses are defined the same as the general RPN learning for detection.

$$L = \lambda_1 L_{rep} + \lambda_2 L_{att} + \lambda_3 L_{det} \tag{11}$$

Using $L$ we train our system following the self-training scheme [44, 43]. Further details can be found in Sec:B.1.

# 4 Experiment

## 4.1 Datasets

We evaluate our methods against various baselines across three different datasets, such as KITTI [6], NuScenes [1], and Waymo [33]. KITTI contains 7481 frames of point clouds for training and validation, and all the data is collected with 64-beam Velodyne LiDAR. NuScenes dataset contains 28130 training and 6019 validation point clouds collected with a 32-beam roof LiDAR. Waymo dataset contains 122000 training and 30407 validation frames of point clouds collected with five LiDAR sensors, i.e., one 64-beam LiDAR and four 200-beam LiDAR.

## 4.2 Implementation Details

For a fair comparison with existing domain adaptive 3D detection methods, we build our model on two base detectors, Second IoU [42] and PointPillars [12], following [44, 43, 17, 8], that are widely used and applicable to most recent detectors with the implementation based on OpenPCDet [35] and parameters from ST3D [44]. Following [17], we first train each detector for 50 epochs with batch-size 8 as a pretraining step using a single NVIDIA A10 GPU. In the self-training stage, we train 30 more epochs for the tuning to adapt to the target domain. The learning rate is set to $1 \times 15^{-4}$ using Adam optimizer with Cosine annealing [47] for scheduling the learning rate. In order to improve the learning stability, the baseline RPN is also learned in addition to the group learning. While the group equivariant RPNs are used for the pseudo label collection during self-training, only the baseline RPN is used for the final testing, ensuring the same execution speed as the existing pipeline. The feature dimensions $d_{bev}$ and $d_{obj}$ for $F_{bev}$ and $\mu$ are both set to 512 for $cAttn$. $\alpha$ for updating the group parameters is empirically set to 0.8. $\lambda_1$, $\lambda_2$, and $\lambda_3$ are set to 0.5, 0.5, and 1.0, respectively.

## 4.3 Comparing Methods

We compare recent existing 3D domain adaptive detection methods, such as SN [38], 3D-CoCo [46], ST3D [44, 43], GPA-3D [17] and DTS [8] with our proposed method. As our method is based on the self-training, we set ST3D [44] as our baseline and show experimental results by comparing with more recent methods. Additionally, we also illustrate the performance of the **oracle** models, which refer to a fully-supervised model on the target domain directly as an upper bound. Following the most recent works [17, 8], all methods are compared in three adaptation scenarios focusing on "car" class: (1) Waymo → NuScenes (2) NuScenes → KITTI (3) Waymo → KITTI.

## 4.4 Evaluation Metric

Following [44, 43, 17, 8], we adopt Average Precision(AP) as our primary evaluation metric and evaluate our model on Bird-Eyes-View (BEV) IoU, $AP_{\text{BEV}}$, and 3D Box IoU, $AP_{\text{3D}}$, with 40 varying recall points and 0.7 as the IoU threshold.

### 4.4.1 Quantitative Result

**Waymo → KITTI** Table 1 (first task) shows the quantitative results of 3D detection in $AP_{\text{BEV}}$ and $AP_{\text{3D}}$. When using Second-IoU as detector [42], our proposed method outperforms the baseline ST3D series for 4.75/11.88 in $AP_{\text{BEV}}/AP_{\text{3D}}$, respectively. Compared with the SOTA method, DTS [8], 1.14/2.21 improvements are made. When using PointPillars as the base detector [12], our approach gains 2.34/3.91 improvements compared to the best performing method, GPA-3D [17].

**NuScenes → KITTI** As shown in Table 1 (second task), our approach shows 0.97/5.80 performance gains in terms of $AP_{\text{BEV}}/AP_{\text{3D}}$ with Second-IoU [42] as the base detector. Compared with SOTA DTS [8], 0.07/1.60 improvements are acquired. With PointPillars [12] as the base detector, our approach exceeds the baseline and DTS by 21.49/41.74 and 2.39/1.04, respectively.

**Waymo → NuScenes** Table 1 (third task) illustrates the adaptation results. Our approach outperforms the baseline and the best-performing method by 8.55/5.72 and 3.27/2.91 in $AP_{\text{BEV}}/AP_{\text{3D}}$, respectively, with Second-IoU as the detector. Similarly, with PointPillars as the detector, 14.89/7.84 and 32.8/1.94 improvements are gained compared with the baseline and SOTA in terms of $AP_{\text{BEV}}/AP_{\text{3D}}$.

Table 1: Quantitative comparisons of the recent domain adaptive 3D detection methods on three adaptation scenarios. The top-3 performing methods are labeled in different colors.

| Task | Methods | SECOND-IOU | | PointPillars | |
|---|---|---|---|---|---|
| | | $AP_{BEV}/AP_{3D}$ | Closed Gap | $AP_{BEV}/AP_{3D}$ | Closed Gap |
| Waymo → KITTI | Source Only | 67.64/27.48 | - | 47.8/11.5 | - |
| | SN [38] | 78.96/59.20 | 72.33/69.00 | 27.4/6.4 | -55.14/-8.49 |
| | 3D-CoCo [46] | - | - | 76.1/42.9 | 76.49/52.25 |
| | ST3D [44] | 82.19/61.83 | 92.97/74.72 | 58.1/23.2 | 27.84/19.47 |
| | ST3D++ [43] | 80.78/65.64 | 83.96/83.01 | - | - |
| | GPA-3D [17] | 83.79/70.88 | 103.19/94.41 | 77.29/50.84 | 79.70/65.46 |
| | DTS [8] | 85.80/71.50 | 115.9/95.7 | 76.1/50.2 | 76.50/64.4 |
| | Ours | 86.94/73.70 | 123.2/100.4 | 78.44/54.11 | 82.81/71.0 |
| | Oracle | 83.3/73.5 | - | 84.8/71.6 | - |
| NuScenes → KITTI | Source Only | 51.8/17.9 | - | 22.8/0.5 | - |
| | SN [38] | 59.7/37.6 | 25.1/35.4 | 39.3/2.0 | 26.6/2.1 |
| | 3D-CoCo [46] | - | - | 77.0/47.2 | 87.4/65.7 |
| | ST3D [44] | 75.9/54.1 | 76.6/59.5 | 60.4/11.1 | 60.6/14.9 |
| | ST3D++ [43] | 80.5/62.4 | 91.1/80.0 | - | - |
| | DTS [8] | 81.4/66.6 | 94.0/87.6 | 79.5/51.8 | 91.5/72.2 |
| | Ours | 81.47/68.2 | 98.3/90.0 | 81.89/52.84 | 95.3/73.6 |
| | Oracle | 83.3/73.5 | - | 84.8/71.6 | - |
| Waymo → NuScenes | Source Only | 32.91/17.24 | - | 27.8/12.1 | - |
| | SN [38] | 33.23/18.57 | 1.69/7.54 | 28.1/12.98 | 2.41/4.58 |
| | 3D-CoCo [46] | - | - | 33.1/20.7 | 25.00/44.79 |
| | ST3D [44] | 35.92/20.19 | 15.87/16.73 | 30.6/15.6 | 13.21/18.23 |
| | ST3D++ [43] | 35.73/20.90 | 14.87/20.76 | - | - |
| | GPA-3D [17] | 37.25/22.54 | 22.88/30.06 | 35.47/21.01 | 36.18/46.41 |
| | DTS [8] | 41.2/23.0 | 43.7/32.80 | 42.2/21.5 | 67.9/49.0 |
| | Ours | 43.84/24.42 | 57.56/40.66 | 44.31/22.15 | 77.88/52.34 |
| | Oracle | 51.9/34.9 | - | 49.0/31.3 | - |

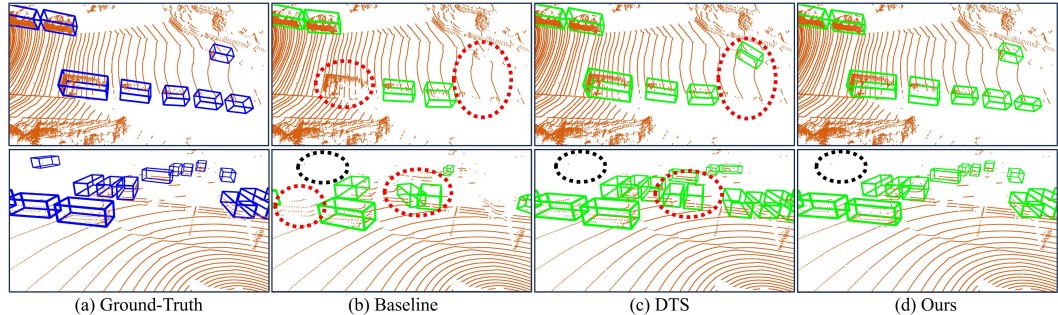

| (a) Ground-Truth | (b) Baseline | (c) DTS | (d) Ours |

Figure 4: Qualitative comparison of Baseline ST3D [44], DTS [8] and ours on NuScenes to KITTI adaptation scenario (top) and Waymo to NuScenes(bottom) adaptation scenarios.

### 4.4.2 Qualitative Result

Figure 4 compares the 3D detection results of the baseline [44], DTS [8] and our methods. Due to the conservative pseudo-label selection policy and absence of methods addressing variations in object size or foreground density, the baseline struggles to detect objects with comparably less common sizes or further away with different densities (red circles in 1st row (b)). Moreover, some dominant object shapes make the detector overfit, leading to false positive detection of road structure (red circles in 2nd row (b)). DTS [8] improves the inter-domain density variance problem presented in the baseline. However, it still encounters the overfitting problem to certain geometric shapes in the source domain, leading to the same false positive detection of the road structure as the baseline (red circles in 2nd row (c)). Moreover, despite the training for foreground density invariant, the objects appearing sparse due to the distance remain false negative (red circles in 1st row (c)) in DTS. The observation suggests that another factor, in addition to the foreground density, causes the inter-domain gap. On the other hand, our proposed method improves both false positive and negative detections, demonstrating a more robust adaptation ability than DTS, which is the best-performing method.

Table 2: Impact of each component in $AP_{\text{BEV}}$ and $AP_{\text{3D}}$ on Waymo to NuScenes adaptation.

| | Group | $L_{att}$ | $L_{repel}$ | Exp. Up. | $AP_{\text{BEV}}$ | $AP_{\text{3D}}$ |
|---|---|---|---|---|---|---|
| (a) | | | | | 35.92 | 20.19 |
| (b) | ✓ | | | | 39.48 | 22.66 |
| (c) | ✓ | ✓ | | | 40.82 | 23.14 |
| (d) | ✓ | | ✓ | | 41.14 | 23.61 |
| (e) | ✓ | ✓ | ✓ | | 41.64 | 24.55 |
| (f) | ✓ | ✓ | ✓ | ✓ | **44.47** | **25.91** |

Table 3: Impact of grouping methods and $\alpha$ on NuScenes to KITTI adaptation.

| | Proximity-based | | GMM-based | |
|---|---|---|---|---|
| | $AP_{\text{BEV}}$ | $AP_{\text{3D}}$ | $AP_{\text{BEV}}$ | $AP_{\text{3D}}$ |
| $\alpha$=0.4 | 79.82 | 64.77 | 81.37 | 66.95 |
| $\alpha$=0.6 | 81.02 | 65.63 | 82.29 | 67.52 |
| $\alpha$=0.8 | 81.61 | 66.20 | **82.78** | **67.93** |
| $\alpha$=0.99 | 81.95 | 66.83 | 81.80 | 66.86 |

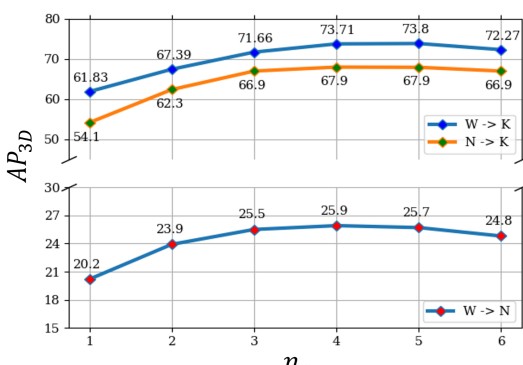

Figure 5: Impact of $n_{\text{g}}$ on three adaptation scenarios in $AP_{\text{3D}}$. Here $W \rightarrow K$, $N \rightarrow K$, and $W \rightarrow N$ refer to Waymo→KITTI, NuScenes→KITTI, and Waymo→NuScenes adaptations.

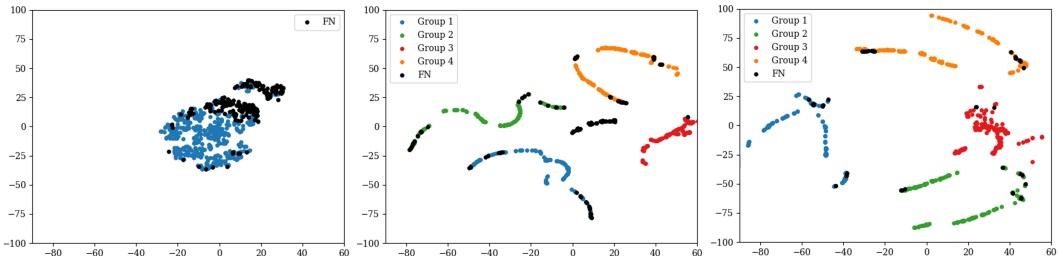

Figure 6: Comparison of DTS [8] (left), ours without explorative update (middle), and ours with explorative update (right) in Waymo → NuScenes with t-SNE visualization. Here, the foreground features are extracted using ground-truth boxes using $F_{\text{bev}}$ for DTS and $F_{\text{cbev}}$ for ours.

### 4.5 Ablations

**Impact of Each Component** Table 2 shows the impact of each component in Waymo→NuScenes adaptation using Second IoU [42] in terms of $AP_{\text{BEV}}/AP_{\text{3D}}$. For this experiment, we progressively add each core component to the baseline (a) to see how they affect the performance. As grouping ensures every group has similar attention during training and prevents only a few dominant object types from having high confidence scores, it significantly improves the performance 9.91%/12.23% from the baseline. From this result, $L_{att}$ (c) and $L_{repel}$ (d) improve 3.39%/2.11% and 4.20%/4.19% respectively, suggesting that making the groups distinctive has more impact than making the intra-group cohesive. This is expected because, during the group-equivariant RPN training, samples inside each group are already learned to be cohesive against background and samples from other groups, indirectly supervised to be close to each other. Nevertheless, when combined in (e), $L_{att}$ and $L_{repel}$ improves 15.92%/21.59% from the baseline, demonstrating the synergy of the two losses. On top of this, the explorative update considerably boosts the performance by 23.80%/28.33%, proving that using samples found in target (pseudo-labels) for the update further reduces the inter-domain gap, as can also be seen in Figure 6 (a group of FN in the middle disappears in the right figure).

**Impact of $n_g$** is illustrated in Figure 5 in $AP_{\text{3D}}$ for all adaptation scenarios presented in the main experiments. For this experiment, we include all components with the same hyper-parameters and only change $n_g$ to see the impact. As can be seen, increasing $n_g$ improves the performance, reaching the top around when $n_g$ is set to 4. In principle, having many groups would make the group-equivariant RPN discover more object types while ensuring the same attention for each group during learning. However, in practice, the performances start decreasing when $n_g$ is set around 6, suggesting that having too many groups increases the risks of overfit problems due to a small number of objects in each group or underfit problems due to high complexity for learning $F_{cbev}$. We also find that the training result is most stable when $n_g$ is set to 3.

**Effect of GMM based Grouping and $\alpha$** are illustrated in Table 3 to compare: (1) Proximity-based grouping and (2) GMM-based grouping with varying $\alpha$. For the proximity-based grouping, we discard $\sigma$ and $\phi$ from GMM, and determine the group of each sample as the closest $\mu$ from the

samples using $L2$ distance while all other configurations stay the same. As can be seen, GMM-based grouping constantly shows better performances in both $AP_{\text{BEV}}/AP_{\text{3D}}$ in all $\alpha$. This is due to $\sigma$ and $\phi$ that preserve the characteristics of each group more compared to only $\mu$ based on previously seen samples, improving the stability during learning with similar objects grouped together. Additionally, setting $\alpha$ too large or small degrades the performances, as small $\alpha$ discourages the explorative update, while too large $\alpha$ could result in instability as the input $\mu$ for training RPN constantly changes more. Also, setting $\alpha = 0.6$ results in 0.49/0.66 higher in $AP_{\text{BEV}}/AP_{\text{3D}}$ compared to setting $\alpha = 0.99$, suggesting the advantage from exploration surpasses the stability. Interestingly, unlike GMM-based grouping, setting higher $\alpha$ constantly shows increasingly better performance with the proximity-based grouping, proving inherent instability without $\sigma$ and $\phi$. Nevertheless, in nearly all the configurations of $\alpha$, the GMM-based grouping persistently outperforms the best-performing method, DTS [8].

## 5    Conclusion and Limitations

In this paper, we present *GroupExp-DA* that learns object groups, which can be used to bridge the inter-domain gap with (1) less bias by the dominant objects in the available label sets and (2) consideration of multiple factors for creating inter-domain gaps in a data-driven manner. This is achieved by utilizing the group equivariant spatial features that connect the group feature and spatial features to be learned together with the existing detection loss function. Nevertheless, all methods, including ours, struggle to detect objects with extremely sparse foreground points (black circles in 2nd row), as shown in Figure 4, because those objects do not contain distinctive features to being well-learned as groups due to extreme sparsity, which remains as one of the challenges to be explored.

## Acknowledgments

This work was supported by Innovate UK Grant No: 10107189 "ScanSpot: 3D-Modelling "Digital Twin" Data For New Insurance Products".

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

# Appendix

## B    Supplementary Material

In this supplementary material, we provide the following additional information:

- Details of Self-Training Implementation
- Additional Qualitative Results
- Multi-Class Adaptation Results

### B.1    Details of Self-Training Implementation

In the self-training framework [44], a batch consists of an identical number of point cloud scans and labels from the source and target domains to train the base 3D detector. For training, the ground-truths from source domain and pseudo-labels from the target domain are considered the same. For each cycle of the self-training, we train our system for 2 epochs using $L$ in Eqn. 11 and collect the pseudo-label sets to merge into $Y$ using the model's inference. Here, the collection of the pseudo-labels takes place in the training set of the target domain. Utilizing the new $Y$, we progressively train the detector for $n_{se}$ epochs again and iterate the same process. The confidence score of the detection to be considered as the pseudo label is set to 0.5, and $n_{se}$ is set to 2 epochs.

### B.2    Additional Qualitative Results

Figure II shows additional qualitative results on Waymo→KITTI and Waymo→NuScenes. For all the qualitative results of previous works, we only include the results that reproduce the reported results on their papers based on the published source codes for each dataset at the time of submission

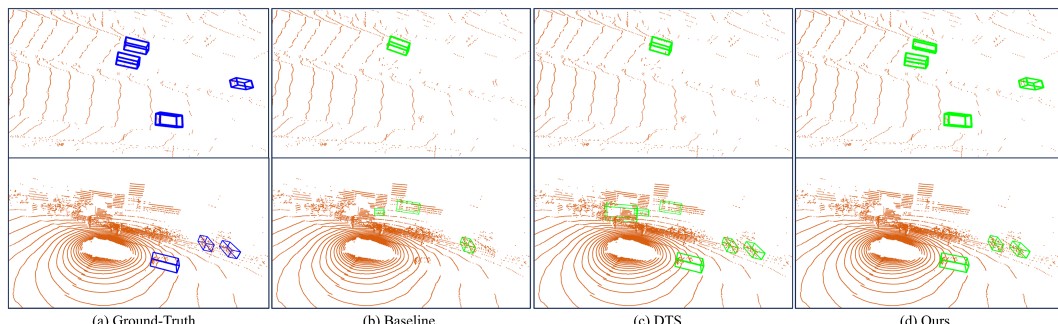

| (a) Ground-Truth | (b) Baseline | (c) DTS | (d) Ours |

Figure II: Qualitative comparison of Baseline ST3D [44], DTS [8] and ours on Waymo to KITTI adaptation scenario (top) and Waymo to NuScenes(bottom) adaptation scenarios.

## B.3   Multi-class Adaptation

Table II shows the result of multi-class adaptation. As our method is designed as a simple add-on that can be applied to general existing pipelines, we apply our pipeline on top of the multi-class adaption approach, REDB [3]. As can be seen, our GroupEXP-DA and REDB show synergy, improving the performance further.

| Method | Car $AP_{BEV}/AP_{3D}$ | Pedestrian $AP_{BEV}/AP_{3D}$ | Cyclist $AP_{BEV}/AP_{3D}$ |
|---|---|---|---|
| Source Only | 39.15/7.65 | 21.54/16.87 | 6.31/2.44 |
| ST3D [44] | 71.50/48.09 | 22.64/17.61 | 7.86/5.20 |
| ReDB [3] | 74.23/51.31 | 25.95/18.38 | 13.82/8.64 |
| Ours (SA) | 72.39/50.01 | 24.47/17.80 | 10.90/6.81 |
| Ours+ReDB | 75.12/52.18 | 26.47/18.51 | 14.92/9.13 |
| Oracle | 83.29/73.45 | 46.64/41.33 | 62.92/60.32 |

Table II: Comparison with multi-class adaptation setting for NuScenes → KITTI adaptation task. Here, Ours (SA) refers to the naive extension of single-class adaptation to multi-class adaptation, and Ours + ReDB stands for the proposed method added on top of ReDB. Here, all of ours use three groups.

