# OpenReview forum: "Towards Learning Group-Equivariant Features for Domain Adaptive 3D Detection"
_NeurIPS.cc/2024/Conference — NeurIPS 2024 poster_

### Official Review · Reviewer_mXQC · 2024-07-06

**Soundness:** 3
**Presentation:** 2
**Contribution:** 2
**Rating:** 5
**Confidence:** 3

**Summary:**

In this paper, the authors utilize a Gaussian Mixture Model (GMM) based grouping & exploration module on the object descriptors extracted from foreground points, then fed the group features to the Group-Correlation module and fuse with RPN to selectively detects objects similar to the individual group, which includes group-equivariant spatial feature to ensure equal learning attention for different object groups and explorative group update strategy to reduce false negative detections by redistributing available labels. The paper conducts domain adaptive 3D detection experiments on KITTI, NuScenes, and Waymo datasets, and ablation studies on the impact of each component, the impact of $n_g$, and the effect of GMM based Grouping and $\alpha$.

**Strengths:**

Originality: Apply GMM based grouping in domain adaptation for 3D object detection, which differs from previous methods.
Quality: Compare to many previous methods, and achieve good performance. Good ablation studies.
Clarity: Images are clear and easy to understand.
Significance: Provide idea that use GMM for Group-Equivariant Features Learning in Domain Adaptive 3D Detection.

**Weaknesses:**

Lack of implementation details and the code is not provided, for example the detailed parameters and hyper-parameters of the Models, like the voxel size. It's also better to provide the model size or runtime of the model, to make sure the performance gain is not from the increment of the model size, and to make sure the model have an apple-to-apple comparison with the previous methods.

Lack of variety of the object categories, such as pedestrians, cyclists, and so on. The experiments in the paper mainly focuses on cars, it's better to include other types of objects such as pedestrians, cyclists, and so on. In addition, lack the variety of the visualization in the qualitative results.

Minor:
In References, the authors of [39] is missing.

**Questions:**

Have you tried different 3D detectors like CenterPoint and other recent SOTA 3D detectors? Is the performance gain of the proposed method still significant when the baseline is high? If it is, and even achieves SOTA, it will be helpful to prove the significance and the contribution of the proposed method.

**Limitations:**

Yes.

---

> ### Author Rebuttal · Authors · 2024-08-07
>
> We thank the reviewer for the constructive assessment of our work. In the following, we address the concerns point by point. Please feel free to use the discussion period if you have any additional questions.
>
> 1) Lack of variety?
> We address the multi-class adaptation in the main author rebuttal. In addition, as you suggested, we uploaded additional visualizations and performance with other base detectors in Figure A2 and Table A3.
>
> 2) What are the voxel size, model size, and running speed?
> The voxel size is set following the comparing methods ST3D and DTS, which depend on the adaptation task. Please refer to Table A4 in the rebuttal pdf for the model size and running speed.

---

> ### Author Response · Authors · 2024-08-07
>
> We would like to add more discussion on a few more details, as official comments are available for discussion during the whole author-reviewer discussion phase. Here are a few more details that we wish to clarify based on the reviewer's questions and concerns.
>
>
> **3) Have you tried different 3D detectors like CenterPoint and recent SOTA detectors? Did the proposed model go through apple to apple check?**\
> In the main paper, we have reported the result with mainly Second-IoU [37] and PointPillars [9] as the base detectors for the fair comparisons with the existing works, where all the configurations, such as voxel size, learning rate, number of filters, thresholds, etc, for the adaptation stay the same, as mentioned in Sec. 4.2 of the main paper.
> As one of the other famous base detectors, we have included additional results with CenterPoint, as the reviewer suggested, and DSVT[32] in Table A3 for the NuScenes to KITTI adaptation task. We selected DSVT as one of the most recent base detectors due to the fact that it is designed for LiDAR only, which is compatible with our purpose and is one of the top-performing methods.
> Specifically, Table A3 shows the apple-to-apple comparison of DTS and our proposed method given the same base detectors, CenterPoint and DSVT backbone, followed by CenterPoint Head.
> Interestingly, using CenterPoint Head leads to worse results compared to anchor-based methods, such as Second-IoU, when we directly use the model only trained on the source domain to target without self-training.
> We think this is because the anchor-based methods inherently encourage the spatial voxel features (F_{bev}) to learn the bigger surrounding context with supervision to predict IoU between boxes, rather than center position-based supervision from CenterPoint.
> Nevertheless, our proposed method constantly outperforms the SOTA method for domain adaptive detection, DTS, under the same setting as the base detectors. Also, changing the base detectors to the more recent method, such as DSVT, leads to more improvement for adaptation with around 1.89 gain in AP_{3D}, proving the applicability of our method.
>
>
>
>
> **4) Performance Gain by model size?**\
> Table A4 shows the runtime analysis for inference speed and model size on an NVIDIA A10 GPU. Here, on KITTI validation set, we measured the inference speed of three models, namely ST3D, DTS, and ours. As can be seen, the current SOTA DTS requires around 43% more memory while underperforming than GroupExp-DA, suggesting that the improvement made by our proposed method is not simply based on the increment of model size.
> The inference speed of our method is among the slowest. However, we would like to note that the speed has comparably more room for improvement than the precision without sacrificing model size by using optimized software such as onnxruntime, tensorRT, etc.
>
>
>
> **5) More discussions on multi-class adaptation settings.** \
> We have conducted comprehensive experiments designed for other categories of objects, such as pedestrians and cyclists. This can be found in the main author rebuttal for all reviewers. As you kindly suggested, we added more visualizations focusing on cyclists and pedestrians in KITTI validation set, which can be found in Figure A2.
>
>
>
> **6) Code is not provided.**\
> We will release the code and checkpoints as soon as the paper gets accepted.
>
> **7) Missing authors of [39].**\
> Thank you for letting us know about the reference issue. We will fix it in the final version.
>
>
> Again, we thank the reviewer for constructive and valuable comments on our paper. We will make sure to update the main paper based on the additional results acquired based on your comments. Please let us know if you have any further comments or questions.

---

> ### Author Response · Authors · 2024-08-12
>
> Dear Reviewer mXQC,
>
> We sincerely appreciate your efforts in providing constructive and professional reviews. During the rebuttal period, we have tried our best to provide feedback and rebuttal to address your concerns. All of them will be involved in our final revised version.
>
> If you have any further concerns, we are more than happy to discuss more before the author-reviewer interactive discussion period ends.
> Thank you again for your insightful and meaningful comments.
>
> Best regards,
> Authors of Paper 9774

---

> > ### Comment · Reviewer_mXQC · 2024-08-13
> >
> > Thank you for the detailed rebuttal and the additional experiments provided in Table A3 and A4. These results address some of my concerns. I appreciate your commitment to including these findings in the final paper. Based on the improvements and the clarifications provided, I will adjust my score to borderline accept. Please ensure that these experimental results are incorporated into the final version of the paper.

---

> > > ### Author Response · Authors · 2024-08-13
> > >
> > > Dear Reviewer mXQC,
> > >
> > > We are delighted to hear that our responses to your concerns have contributed to a positive assessment of our work. As you suggest, we will make sure to include the content in the rebuttal to the final version of the paper.
> > > Please let us know if you have any further questions.

---

### Official Review · Reviewer_5ZUx · 2024-07-11

**Soundness:** 3
**Presentation:** 3
**Contribution:** 3
**Rating:** 6
**Confidence:** 4

**Summary:**

The performance of 3D object detection in large outdoor point clouds suffers in unseen environments due to inter-domain gaps. Existing domain adaptation methods, focusing on single factors like object size or shape, still leave substantial gaps. This work proposes a grouping-exploration strategy framework to address biased pseudo-label collection and multiple contributing factors. The framework clusters label sets to ensure balanced learning and introduces an explorative group update strategy to reduce false negatives. This method, applicable as a simple add-on to existing detectors, significantly bridges inter-domain gaps in datasets like NuScenes, Waymo, and KITTI.

**Strengths:**

+ The presentation is commendable (Figures 1-4 are particularly well-executed), and the writing is clear and easy to follow.
+ A keen observation is highlighted in Figure 1: the baseline adaptation method mainly detects objects with features near the mean of the distributions, indicated by red circles.
+ The proposed explorative update is well-motivated, aiming to cover a broader range in the target domain rather than baised to the source.

**Weaknesses:**

One of the baselines, the performance of REDB, is not fairly reported. REDB addresses more realistic scenarios where the 3D detector is trained and adapted for all object categories simultaneously. However, in Table 1, the adaptation focuses only on a single category: car. Therefore, reporting the performance of REDB in Table 1 is inappropriate. Additional experiments under a multi-class setting are necessary to compare the proposed GroupExp-DA with REDB, validating the effectiveness of GroupExp-DA in more realistic scenarios.

**Questions:**

Regarding the weakness, could you provide the results for different adaptation tasks under the multi-class adaptation setting?

**Limitations:**

The limitation has been discussed, but potential solutions have not been provided.

---

> ### Author Rebuttal · Authors · 2024-08-07
>
> We thank the reviewer for the constructive assessment of our work.
>
> We address the concern you raise in the main author rebuttal. In particular, Table A2 shows the experimental results for the multi-class adaptation setting.
>
> Please feel free to use the discussion period if you have any additional questions.

---

> ### Author Response · Authors · 2024-08-08
>
> We would like to add more discussion on a few more details, as official comments are available for discussion during the whole author-reviewer discussion phase. Here are a few more details that we wish to further clarify based on the reviewer's questions and concerns.
>
> **1) Multi-class adaptation experiment and ReDB.**\
> As mentioned in the main rebuttal for all reviewers, Table A2 shows performances of ST3D, ReDA, GroupExp-DA without ReDA, and GroupExp-DA with ReDA for NuScenes to KITTI adaptation. For a fair comparison, we brought the result reported from the ReDB paper. For ours with ReDA implementation,
> we simply add cross-domain examination, diversity sampling, and class-balanced self-training available from the GitHub repository of ReDB, which are proposed for multi-class adaptation settings. Other configurations also follow ReDB's configuration using Second-IoU as the base detector.
> Interestingly, for ReDB + GroupExp-DA, we observe that the performance improvement from the ReDB model is more significant for 'cyclist', which is 8\% and 5.67\% for $AP_{BEV}$ and $AP_{3D}$, compared to 1.1\% and 1.7\% for 'car', suggesting a potential synergy of the two methods for more challenging class, cyclist.
> The straightforward explanation would be that the focuses of ReDB and GroupExp-DA are different in making improvements, which leads to additional performance boosts when they are both used together for improving different aspects. We are further investigating this to get more intuition.
>
> **2) New Table for multi-class adaptation setting.**\
> We agree with the reviewer's point of view that ReDB needs to be compared differently from other baselines that focus on the single-class adaptation setting. We will separate ReDB from the current Table 1. Specifically, we will create another table focusing on the multi-class adaptation setting with proper performances of other baseline methods from the ReDB paper as a more challenging yet closer to real-world scenarios. Table 1 will remain for performance comparison with the single-class adaptation-based methods without ReDB.
>
>
>
> We appreciate the reviewer for the constructive and valuable comments on our paper. We will make sure to update the main paper based on the additional results acquired from your comments. Please let us know if you have any further comments or questions.

---

> > ### Comment · Reviewer_5ZUx · 2024-08-11
> >
> > Thanks for the response. Most of my concerns are addressed. I will raise to positive rating.

---

> > > ### Author Response · Authors · 2024-08-11
> > >
> > > Dear Reviewer 5ZUx,
> > >
> > > We are pleased to hear that our responses to your questions have been well received and contributed to a positive assessment of our work. Please let us know if you have any further questions.

---

### Official Review · Reviewer_mHQJ · 2024-07-13

**Soundness:** 2
**Presentation:** 2
**Contribution:** 2
**Rating:** 5
**Confidence:** 4

**Summary:**

This paper deals with the domain adaptive 3D detection based on point clouds, where labels are available for source domain and not for target domain. The authors noticed that previous works tried to model and minimize the domain gap in terms of one specific factor, like shape, density, etc., and proposed to cluster the features into different groups. During training, the cross-attention is performed between conventional point feature and the group feature to perform the group-aware detection. The authors combine the proposed method with the conventional self-training pipeline and empirically show that the proposed method can improve the domain adaptive detection performance.

**Strengths:**

- The paper is generally easy to follow and understand.

- The idea is generally reasonable.

- Experiment results are good.

**Weaknesses:**

- In my understanding, the proposed method mainly enhances the point representations with combing the group feature (i.e., fusing the representations of similar points). As the authors discussed in the abstract and introduction, self-training may generate biased pseudo-label collection. I just wonder how the proposed method correct such a bias in self-training.

- Some technical details are not clear enough. For example, is $\mu$ directly updated by gradient back-propagation? Or is it calculated based on current features and updated using moving average? What is the "closed gap" in Table 1?

- It is better to summarize the training process into an Algorithm table to help the reader better understand the details and pipeline of proposed framework.

- What is the effect of $\lambda_1$, $\lambda_2$ and $\lambda_3$ in Eq. (11)? How to choose them in practice? Lack hyper-parameter study.

**Questions:**

See the weakness part.

**Limitations:**

The authors adequately addressed the limitations.

---

> ### Author Rebuttal · Authors · 2024-08-07
>
> We thank the reviewer for the constructive assessment of our work. In the following, we address the concerns point by point. Please feel free to use the discussion period if you have any additional questions.
>
> (1) How does the proposed method correct the bias in self-training?
> We would like to let you know that we address the concern in the author rebuttal. The grouping and explorative group update encourage objects with similar features to be close to each other in the feature space. This leads each group to focus on objects with different features, whereas previous works are prone to be biased by objects with globally dominant features, as they are similar to having one group. As the self-training is designed to collect pseudo-label using highly confident detections, many groups with different focuses are able to find larger variation of objects with different features, alleviating the bias problem. Figure A1 demonstrates the concept on NuScenes. Here, we extract each object feature using ground truth object boxes with RoI pooling from $F_{bev}$ for DTS and $F_{cbev}$ for ours. The left figure shows that for all methods, the recall improves for the object with features similar to those of the group center.
> The figures in the middle and right show that multiple groups are able to discover more objects with variety.
>
> (2) Hyperparameter selection is addressed in the main author rebuttal.
>
> (3) Training pipeline is added in the rebuttal pdf that we uploaded.

---

> ### Author Response · Authors · 2024-08-08
>
> We would like to add more discussion on a few more details, as official comments are available for discussion during the discussion phase. Here are a few more details that we wish to further clarify based on the reviewer's questions and concerns.
>
> **1) Clarification on overcoming the bias in self-training and Figure A1.**\
> As mentioned in the rebuttal, our key insight is that each group distributes the focus of RPN to objects with different features, side-stepping the bias caused by globally dominant features of objects, as can be seen in Figure 1 of the main paper.
> In order to further demonstrate this bias problem, we attempt to show that objects in the target domain with similar features as a dominant feature in the source domain are likely to be detected more and used as pseudo labels, causing bias in self-training as follows.
> To find the dominant feature and compare the similarity between the dominant feature and each object, we first acquire the object feature using the spatial feature, which is $F_{bev}$ for DTS and $F_{cbev}$ for ours, with RoI Pooling using ground-truth boxes in the source domain. The dominant feature is then acquired by averaging all the
> object features, assuming that outnumbering dominant features will naturally be reflected in the mean. Figure A1 (left) shows recall with respect to the distance between the dominant feature from the source domain and each object feature in the target domain (NuScenes) after self-training. As can be seen, it is clear that all methods
> exhibit higher recall for the objects that are closer to the dominant features, the centers acquired by averaging, and the recall decreases for objects that have less similar features to the dominant features. Here, similar to the source domain, we acquired the object feature
> in the target domain by utilizing the ground-truth boxes and spatial features for the analysis.
> Figure A1 (middle) shows the number of objects depending on the similarity between the center and object feature in the target domain. Based on the observation from Figure A1 (left), we can conclude that having more number of objects close to the center in Figure A1 (middle), such as R1 and R2 in x-axes, means the detector is less biased.
> As can be seen, our grouping strategy leads to more samples in R1 and R2, suggesting that the proposed GroupExp-DA trains the base detector with fewer biases compared to the current SOTA, DTS, for domain adaptive detection. Also, the base detector trained with both the grouping and explorative update consistently shows better performance (green bar) than the base detector trained only with the grouping with an update using all available labels (orange bar), highlighting the contribution of the explorative group update for self-training. Figure A1 (right) shows how each group contributed to the formation of the green bar (Ours+Group.+Exp.Up.) in Figure A1 (middle).
>
> **2) Is $\mu$ directly updated by gradient back-propagation?**\
> $\mu$ is indirectly updated by gradient back-propagation. In a batch, it is the mean of object descriptors (MLP + Global max pooling in Sec. 3.2) that belong to the same group. This batch-wise mean $\hat{\mu}$ is linearly combined (moving average) to existing $\mu$ to preserve the gradient flow from the object descriptor extraction module in Sec 3.2 for learning.
>  Therefore, since the group equivariant spatial feature uses $\mu$ for Regional Proposal Network (RPN),
> the object descriptor extraction module is updated using the gradient calculated from $L_{det}$ with additional group contrastive loss $L_{att}$ and $L_{repel}$  to produce better object descriptors, which are used to update \mu. This process is now more clearly summarized in the Algorithm table in the rebuttal pdf, as you kindly suggested. In addition, the contribution of each loss can also be found in Table 2 of the main paper.
>
> **3) What is the closed gap in Table 1?**\
> Closed gap is the metric proposed in ST3D to show how much performance gap is closed between Source Only model, which is trained only on the source domain and tested in the target domain, and Oracle model, which is trained on the target domain with the ground truth.
> It is calculated as follows: Closed Gap = (AP_{model} - AP_{source only}) / (AP_{oracle} - AP_{source only}). Here, AP_{model} is the Average Precision (AP) calculated using the tested model's prediction. Similarly, AP_{source only} and AP_{oracle} are AP predicted using the source-only and oracle models. We will clarify this in the main paper. We appreciate your comment.
>
> **4) Algorithm.**\
> The training pipeline is now summarized into an Algorithm table in the uploaded rebuttal pdf, as you kindly suggested. For the inference, only lines 5,6 and 7 in the table are used in the pipeline.
>
> We are grateful for the reviewer's constructive comments on our paper. We will update the main paper based on the results from your comments. Please let us know if you have any further comments or questions.

---

> > ### Comment · Reviewer_mHQJ · 2024-08-13
> > **Re: rebuttal**
> >
> > Thanks for the authors detailed response. I will raise my rating. Hope the authors could modify their paper accordingly to make it clearer.

---

> > > ### Author Response · Authors · 2024-08-13
> > >
> > > Dear Reviewer mHQJ,
> > >
> > > We are pleased to learn that our responses to your concerns have positively influenced your evaluation of our work. As you suggest, we will make sure to include the content in the rebuttal to the final version of the paper.
> > > Please let us know if you have any further questions.

---

> ### Author Response · Authors · 2024-08-12
>
> Dear Reviewer mHQJ,
>
> We sincerely appreciate your efforts in providing constructive and professional reviews. During the rebuttal period, we have tried our best to provide feedback and rebuttal to address your concerns. All of them will be involved in our final revised version.
>
> If you have any further concerns, we are more than happy to discuss more before the author-reviewer interactive discussion period ends.
> Thank you again for your insightful and meaningful comments.
>
> Best regards,
> Authors

---

### Official Review · Reviewer_CYB3 · 2024-07-13

**Soundness:** 2
**Presentation:** 3
**Contribution:** 3
**Rating:** 6
**Confidence:** 4

**Summary:**

The authors propose a new framework that leverages a grouping-exploration strategy to address the inter-domain gap observed in 3D object detection. The core of their approach is to divide the available labels into multiple clusters, ensuring equal learning attention across these groups using group-equivariant spatial features. This method helps prevent the imbalance caused by dominant object types and reduces false negatives by incorporating an explorative group update strategy during the learning process. The framework is designed to be an add-on that can easily integrate with existing detection systems. The proposed method shows substantial improvements in bridging the inter-domain gaps compared to existing approaches, demonstrated across datasets like NuScenes, Waymo, and KITTI.

**Strengths:**

1. The proposed grouping module and group-region correlation module are intuitive and technical sound.
2. The author provides extensive ablation studies for the proposed group-extraction method.

**Weaknesses:**

My major concern lies in the generality of the proposed method. Since the point cloud of "car" object is large and unique, it's easier to group for cars. Can the proposed method generalize to other categories? I hope the author can further validate the proposed method from this perspective.

**Questions:**

Please refer to the weakness part.

---

> ### Author Rebuttal · Authors · 2024-08-07
>
> We thank the reviewer for the constructive comment on our work.
>
> We would like to let you know that the concern that you raised is addressed in the author rebuttal, which is visible to all reviewers.
> In particular, Tables A1 and A2 in the uploaded rebuttal pdf show the experimental results for other categories for NuScenes $\rightarrow$ KITTI adaptation tasks. Please feel free to use the discussion period if you have any additional questions.

---

> ### Author Response · Authors · 2024-08-08
>
> We would like to add more discussion on a few more details, as official comments are available for discussion during the whole author-reviewer discussion phase. Here are a few more details that we wish to further clarify based on the reviewer's questions and concerns.
>
> **1) Generalization ability to other categories.**\
> As mentioned in the main rebuttal, we have additionally conducted comprehensive experiments for other categories, such as pedestrians and cyclists, following the reviewer's comments, which are available in the uploaded PDF. Initially, we focused on only 'Car' category, as it is still challenging for domain adaptative detection that most existing works on single-class adaptation focus on.
> As can be seen in both Tables A1 and A2, the proposed GroupExp-DA shows consistent improvements from the baselines, demonstrating its ability to generalize to other categories.
> Although the APs are still the lowest among all categories, we observe an interesting performance improvement from the current SOTAs for 'Cyclist' category, which are 4.58/%/7.06% for the single class adaptation and 8\% and 5.67\% for the multi-class adaptation settings.
> We think this might be because each group in GroupExp-DA focuses on objects with different features, which are beneficial for generalization when the number of available labels in the source domain is significantly smaller.
> For example, in NuScenes, the number of labels for 'cylist' class, which are the sum of 'bicycle' and 'motorist,' is 11,538, while 'car' and 'pedestrian' have 27,449 and 22,447 labels, respectively.
> The imbalance is more significant in KITTI where the number of labels available for 'Cyclist' is 734 while 'Car' and 'Pedestrian' are 14,357 and 2,207, respectively.
>
> We appreciate the reviewer for the constructive and valuable comments on our paper that provided us with new insights. We will make sure to update the main paper based on the additional results acquired from your comments. Please let us know if you have any further comments or questions.

---

> ### Comment · Reviewer_CYB3 · 2024-08-12
>
> Thanks for the strong rebuttal. My concerns have been addressed, and I will raise my rating to weak accept. Please include the content in rebuttal to the next revision.

---

> > ### Author Response · Authors · 2024-08-12
> >
> > Dear Reviewer CYB3,
> >
> > We are pleased to hear that our responses to your questions have been well received and contributed to a positive assessment of our work. We will make sure to include the rebuttal in the final version of the paper. Please let us know if you have any further questions.

---

### Author Rebuttal · Authors · 2024-08-07

Dear Reviewers,

We greatly appreciate your valuable and constructive reviews for our paper. Most of the reviews are very helpful. We thoroughly noted your feedback and will make sure to include all the discussions that we have during the rebuttal in the final version of the paper. We summarized the common questions in the initial reviews as follows. The other questions are answered in detail for individual reviewer's rebuttals.

(1) How well does the proposed system generalize to other categories?
In order to investigate the proposed system's generalization ability, we have conducted additional experiments on other categories, such as pedestrian and cyclist, as suggested. For the comprehensive experiment, we have conducted two experiments: (1) Single-class adaptation (2) Multi-class adaptation. Table A1 shows the performances of the single-class adaptation, which is the primary task of GroupExp-DA. Here, we keep all the parameters the same as the experiment for 'car' and only replace the target category. For both pedestrian and cyclist categories, despite the difference in performance depending on the category, the proposed GroupExp-DA consistently outperforms the current SOTA. We provide a more detailed analysis of the result in the individual reviewer's rebuttal.

Table A2 illustrates the performance of the multi-class adaptation setting. Here, Ours (SA) refers to a naive extension to multi-class, where the same number of groups (=3) is assigned for each category straight-forwardly for learning. On the other hand, Ours + ReDB adopts ReDB on top of ours, which is specialized for multi-class adaptation settings. The experimental results show that GroupExp-DA can be a simple add-on that can be successfully combined with multi-class adaptation methods to improve performance. We describe the result in more detail in the reviewer's rebuttal.

(2) How does the proposed system overcome bias from dominant objects during self-training?

GroupExp-DA overcomes the bias from self-training with grouping and explorative group updates. We first observe that the dominant features of objects in one domain cause inter-domain gaps and lead the detector to be biased because they outnumber other objects, as can be seen in Figure 1 of the main paper. GroupExp-DA attempts to address this imbalance problem by distributing objects into groups. Our key insight is that while grouping, each group finds locally dominant object features that are comparably ignored due to outnumbering globally dominant features in one domain. The learning attention is then equally divided to each group using group-wise object detection, alleviating the imbalance problem. The grouping improves the inter-domain gap because group-wise detection allows for focusing on objects with minor features in one domain, which can contain dominant features in other domains, improving the false negatives for collecting pseudo labels during self-training. Figure A1 in the rebuttal pdf demonstrates the concept in addition to Figures 3 and 6 in the main paper.





(3) How were the hyper-parameters, such as $\lambda$ selected? How sensitive are they?
The hyper-parameters were selected based on the simple principle that the loss for detection and grouping contrastive learning ($\lambda_{1}$, $\lambda_{2}$) should have equal weights. This led us to select the hyper-parameters so that summing up $\lambda_{1}$ and $\lambda_{2}$ equals $\lambda_{3}$. Table A5 show the impact of hyper-parameter change in NuScenes to KITTI adaptation task. In most of the configurations, except for the case where the coefficients are unreasonably low, such as ($\lambda_{1}$=0.1,$\lambda_{1}$=0.1),  GroupExp-DA outperforms the current SOTA, DTS.

---

### Decision · Program_Chairs · 2024-09-25

**Decision:**

Accept (poster)

**Comment:**

Hi,
This draft has been awarded two weak accept and 2 borderline accept. On the basis of positive reviews, this draft is being accepted for publication.


regards

AC